# Heat Stress Effects on Physiological and Milk Yield Traits of Lactating Holstein Friesian Crossbreds Reared in Tanga Region, Tanzania

**DOI:** 10.3390/ani14131914

**Published:** 2024-06-28

**Authors:** Vincent Habimana, Athumani Shabani Nguluma, Zabron Cuthibert Nziku, Chinyere Charlotte Ekine-Dzivenu, Gota Morota, Raphael Mrode, Sebastian Wilson Chenyambuga

**Affiliations:** 1Department of Animal, Aquaculture, and Range Sciences, Sokoine University of Agriculture, Morogoro P.O. Box 3004, Tanzania; asnguluma@yahoo.com (A.S.N.); chenya@sua.ac.tz (S.W.C.); 2SACIDS Africa Centre of Excellence for Infectious Diseases, SACIDS Foundation for One Health, Sokoine University of Agriculture, Morogoro P.O. Box 3297, Tanzania; 3International Livestock Research Institute (ILRI), Nairobi P.O. Box 30709, Kenya; c.ekine@cgiar.org (C.C.E.-D.); r.mrode@cgiar.org; 4Tanzania Livestock Research Institute (TALIRI), Eastern Zone, Tanga P.O. Box 5016, Tanzania; czabronn@gmail.com; 5School of Animal Sciences, Virginia Polytechnic Institute and State University, Blacksburg, VA 24061, USA; morota@vt.edu; 6Animal and Veterinary Sciences, Scotland’s Rural College, Easter Bush, Midlothian EH25 9RG, UK

**Keywords:** lactating Holstein Friesian crossbreds, milk yield, milk composition, physiological parameters, temperature–humidity index

## Abstract

**Simple Summary:**

In lactating dairy cows, as the temperature–humidity index (THI) thresholds increase from moderate to high, the core body temperature, rectal temperature, respiratory rate, and panting score increase, while the milk yield and fat and solid–not–fat percentages decrease. The effects of THI on milk yield, milk composition, and physiological parameters are greater during the hot season and afternoon hours than during the cool season and morning hours. Critical THI thresholds of 77–84 for physiological, milk yield, and milk composition parameters have been established. In general, the lactating Holstein Friesian crossbred dairy cows in the lowland coastal areas of Tanzania experience heat stress (HS), but the crosses with a 75% Holstein Friesian gene level (HF75) are more stressed than those with a 50% Holstein Friesian gene level (HF50). Thus, HF50 is suitable for the warm and humid coastal areas of Tanzania. The findings of this study could help dairy farmers rearing the lactating Holstein Friesian crossbred dairy cows adopt specific HS mitigation strategies to counter the adverse effects of HS at the farm level in the coastal region of Tanzania and other countries with similar climatic conditions.

**Abstract:**

Global warming caused by climate change is a challenge for dairy farming, especially in sub-Saharan countries. Under high temperatures and relative humidity, lactating dairy cows suffer from heat stress. The objective of this study was to investigate the effects and relationship of heat stress (HS) measured by the temperature–humidity index (THI) regarding the physiological parameters and milk yield and composition of lactating Holstein Friesian crossbred dairy cows reared in the humid coastal region of Tanzania. A total of 29 lactating Holstein Friesian x Zebu crossbred dairy cows with 50% (HF50) and 75% (HF75) Holstein Friesian gene levels in the second and third months of lactation were used. The breed composition of Holstein Friesians was determined based on the animal recording system used at the Tanzania Livestock Research Institute (TALIRI), Tanga. The data collected included the daily temperature, relative humidity, daily milk yield, and physiological parameters (core body temperature, rectal temperature, respiratory rate, and panting score). THI was calculated using the equation of the National Research Council. The THI values were categorized into three classes, i.e., low THI (76–78), moderate THI (79–81), and high THI (82–84). The effects of THI on the physiological parameters and milk yield and composition were assessed. The effects of the genotype, the parity, the lactation month, and the interaction of these parameters with THI on the milk yield, milk composition, and physiological parameters were also investigated. The results show that THI and its interaction with genotypes, parity, and the lactation month had a highly significant effect on all parameters. THI influenced (*p* ˂ 0.05) the average daily milk yield and milk fat %, protein %, lactose %, and solids–not–fat %. As the THI increased from moderate to high levels, the average daily milk yield declined from 3.49 ± 0.04 to 3.43 ± 0.05 L/day, while the fat % increased from 2.66 ± 0.05% to 3.04 ± 0.06% and the protein decreased from 3.15 ± 0.02% to 3.13 ± 0.03%. No decline in lactose % was observed, while the solid–not–fat % declined from 8.56 ± 0.08% to 8.55 ± 0.10% as the THI values increased from moderate to high. Also, the THI influenced physiological parameters (*p* ˂ 0.05). The core body temperature (CBT), rectal temperature (RT), respiratory rate (RR) and panting score (PS) increased from 35.60 ± 0.01 to 36.00 ± 0.01 °C, 38.03 ± 0.02 to 38.30 ± 0.02 °C, 62.53 ± 0.29 to 72.35 ± 0.28 breaths/min, and 1.35 ± 0.01 to 1.47 ± 0.09, respectively, as the THI increased from low to high. The THI showed a weak positive correlation with the average daily milk yield and fat percentage, whereas the protein, lactose, and solids–not–fat percentages showed negative relationships with THI (*p* ≤ 0.05). CBT, RT, RR, and PS showed positive relationships (*p* ≤ 0.05) with THI. These negative relationships indicate that there is an antagonistic correlation between sensitivity to HS and the level of production. It is concluded that the THI, the genotype, the parity, and the lactation month, along with their interactions with THI, significantly influenced the milk yield, milk composition, and physiological parameters of lactating Holstein Friesian dairy crosses at THI thresholds ranging from 77 to 84.

## 1. Introduction

Tanzania is ranked as the third largest cattle-producing country in Africa after Ethiopia and Sudan [1]. The country’s cattle population is about 36.6 million [2], of which 96.1% are indigenous breeds [3]. These indigenous breeds have a low genetic potential for milk production. Efforts to improve the milk production of indigenous breeds through crossbreeding with temperate dairy breeds began in the mid-1960s, following the realization that the use of pure exotic dairy breeds was a failure due to a lack of adaptability to the local environment. The crossbreeding of local cattle with temperate dairy breeds has been pursued as a means to increase milk production in the country. Currently, dairy cows comprise only 2.6% of the cattle population in Tanzania [3], and they are mainly the crosses of Holstein Friesian, Jersey, and Ayrshire with the Tanzania shorthorn Zebu breed [1]. The average milk production of these crossbred dairy cows is very low, usually less than nine liters of milk per cow per day [4]. This suboptimal production performance has been attributed in part to the animals’ prolonged exposure to extreme environmental conditions, including high temperatures, humidity, wind speed, cloud cover, and solar radiation [4,5]. It is also due to poor nutrition, management, and production systems [6], in addition to other factors such as the health status and genotype [7,8].

In Tanzania, the dairy cow production system consists of three sub-sectors, i.e., traditional cow-meat-milk, improved smallholder dairy, and commercial dairy farms [9]. The production system in the Tanga region is predominantly a mixed-crop livestock system, involving the cut-and-carry stallfeeding of fodder, forage, maize, and bean crop residues and the supplementation of agro-industrial-by-products [4]. Smallholder dairy farmers keep small herds, less than ten heads of dairy cows per household, consisting of Holstein Friesian and Ayrshire breeds and their crosses [4]. These cattle breeds are predominant among smallholder dairy farmers in Tanzania, especially in the Tanga region [4]. Among the dairy breeds, the Holstein breed is popular among dairy farmers in Tanzania due to its high milk production potential. However, the potential of Holstein cows to emit body heat through skin evaporation is reduced in hot and humid environments. Therefore, Holstein cows are at a higher risk of heat stress (HS) [10]. In recent years, dairy farmers in the Tanga region have identified temperature increases due to climate change as the main problem causing the low productivity and profitability of dairy cows [5].

Tanzania is currently experiencing the negative impacts of climate change, characterized by an average annual increase in temperature of 1.0 °C since 1960 and an average decrease in rainfall of 2.8 mm per month and 3.3% per decade [11]. Increases in ambient temperature, relative humidity, wind speed, and solar radiation above the thermal neutral zone of a dairy cow cause HS [12,13]. Heat stress occurs when the lactating dairy cow experiences a condition or state in which she is unable to dissipate sufficient heat to maintain thermal balance [14,15,16]. The thermoneutral zone for *Bos taurus* dairy cows’ breeds typically ranges from 0.5 °C to 20 °C [17,18]. The exposure of dairy cows to high ambient temperatures and relative humidity alters numerous physiological responses to maintain homeostasis [19]. When physiological mechanisms fail to counterbalance the excessive heat load, dairy cows suffer HS effects [20,21,22]. This is associated with a decline in the feed intake, increased water consumption, a decline in the milk yield [23], alterations in the milk composition and physiological parameters such as the respiratory rate (RR), rectal temperature (RT), core body temperature (CBT), and panting score (PS) [24,25,26,27]. A reduction in the milk yield due to an increase in temperature–humidity index (THI) values has been reported by Ravagnolo and Misztal [24] and West [28]. Ravagnolo and Misztal [24] reported a decline of 0.009 kg and 0.012 kg in milk protein and milk fat percentages, respectively, at a THI threshold above 72 for US Holstein dairy cows. In the Mediterranean climate in Tunisia, Bouraoui et al. [20] reported a milk yield decline of 21% for Holstein Friesian dairy cows. In Kenya, Mbuthia et al. [29] reported that the average milk production loss was −0.29, −0.19, and −0.37 kg/THI unit per day for the first, second, and third lactation, respectively, for Holstein Friesian, Jersey, and Guernsey breeds. In Rwanda, Niyonzima et al. [30] reported that the THI had a negative effect on daily milk production, with a decline of −0.11 kg milk/THI unit at most.

The temperature–humidity index has proven to be a useful tool for measuring the effects of HS in dairy cows, as it uses air temperature and relative humidity with different weighting scales in animals [15,31,32]. Other climatic indices developed to investigate the degree of HS in dairy cows include the adjusted THI, heat load index, thermal stress index, equivalent temperature index, and dairy heat load index [33]. However, these environmental indices defined in the literature have remained largely unexplored in genetic evaluation studies, and THI continues to be the most popular indicator of HS in dairy cows [32,33]. The THI values are generally divided into classes depending on the severity of HS in dairy cows [33]. For instance, Armstrong [12] classified THI < 72 as comfort, 72 < THI < 79 as mild stress, 80 > THI < 89 as moderate stress, and THI > 90 as severe stress. However, the THI thresholds for a dairy cow’s comfort zone vary depending on the production status, acclimatization level, pregnancy status, diet, and climatic conditions such as the wind speed, solar radiation, and relative humidity [17].

The identification of HS thresholds is crucial, as they can be used to monitor a lactating dairy cow’s welfare and implement potential strategies for mitigating HS [34], especially in sub-Saharan climates, where lactating dairy cows are exposed to extreme ambient temperatures and relative humidity [35]. Lactating dairy cows in many parts of sub-Saharan Africa are routinely subjected to high ambient temperatures and relative humidity. However, information on the effect of HS on milk production, milk composition, physiological traits, and income losses to farmers under sub-Saharan climatic conditions is very limited [19]. Moreover, the relationship between a dairy cow’s genotypes and responses to HS has not been established under the smallholder dairy cow’s production system in sub-Saharan countries [19,21]. The objectives of this study were to investigate the effects of HS, as measured by the temperature–humidity index (THI), on the physiological parameters, milk yield, and milk composition of lactating Holstein Friesian crossbred dairy cows reared in the eastern coast of Tanzania. In addition, the study assessed the relationships between the THI and the physiological parameters, milk yield, and milk composition parameters.

## 2. Materials and Methods

### 2.1. Study Site

This study was conducted at the Tanzania Livestock Research Institute (TALIRI)—Tanga dairy cow’s farm located in Tanga municipality. The institute is situated at 5° S and 39° E at an altitude of 6 m above sea level and 6 km inland from the Indian Ocean. The Tanga region is located in the eastern coastal lowlands of Tanzania. The region covers 26,680 km^2^ and lies between latitude 4.965088° S and 5.5743° S and longitude 38.2744° E and 38.7787° E [36]. The coastal lowland has high temperatures and humidities as well as a high heat load, with THIs reaching above 77.29 in the hot season [4]. Additionally, the annual rainfall in the area ranges from 1230 to 1400 mm, falling in two seasons with peaks occurring during April–May and October–November. The mean temperature in cool months (between May and August) is about 20–24 °C and 23–28 °C during the night and day, respectively. The mean temperature ranges between 26 °C and 33 °C, with January and February being the hottest months of the year. The atmospheric humidity of the region ranges between 65% and 100% [3]. Figure 1 shows the mean temperature and THIs during the recording months in hot and cool seasons.

### 2.2. Experimental Farm Characteristics

The TALIRI-Tanga Dairy Farm has 250 dairy cows (of which 107 are lactating) composed of taurine breeds (Holstein-Friesian, Ayrshire, Jersey, and Simmental) and crosses of these breeds with *Bos indicus* breeds (Tanzania Shorthorn Zebu, Boran, and Sahiwal). The Taurine gene levels vary from 50% to 85%. All cows have similar management practices. Breeding is accomplished by natural mating with bulls and artificial insemination. The dairy farm of approximately 15 ha is managed under a semi-intensive production system for milk production. It is an enclosed dairy farm with standard wooden fencing. The dairy farm has eight paddocks of approximately 30 m × 40 m each, which are used for grazing calves and other lactating dairy cows under intensive care. The paddocks are also used for grazing lactating dairy cows during the cool season. It is equipped with trees for cooling dairy cows during grazing. Milking is carried out in the dairy unit, which has a herringbone parlor composed of 12 pens. There is a semi-enclosed free stall barn, approximately 30 m long and 25 m wide, oriented on an east–west axis. The barn is equipped with lying cubicles mixed with sawdust and wood shavings as bedding material. There is a water source around the barn that is used for drinking and cooling lactating dairy cows before milking. The barn is naturally ventilated through the ridge openings. However, there is no special equipment such as fans and sprinklers used to control and modify environmental conditions such as the wind speed, solar radiation, dry bulb temperature, and daily relative humidity, especially in the hot season. There is also a room around the barn that is used to store silage, hay, and maize bran that is fed to lactating dairy cows during milking, especially during the hot season.

The selected lactating Holstein Friesian crossbred dairy cows were between 4 and 6 years old, with an average body weight of 356.18 kg in the hot season and 328.07 kg in the cool season. There is no specific or technical equipment used for milking lactating dairy cows. However, milking is carried out by hand twice a day using small milk cans and buckets of 3–5 L, and the milk yield is measured using cups. After each cow is milked, the milk yield is collected in three milk cans (two of 50 L and one of 40 L) and sold to the restaurant retailers in Tanga town. The average milk yield per day was estimated to be 3.84 and 1.37 L per day for lactating Holstein Friesian crossbred dairy cows in the hot and cool seasons, respectively, which translates to a 305-day lactation milk production of 1171 L and 427 L, respectively. The grazing pasture is composed of hay, Napier grass, straw, *Brachiaria humidicola*, and green forage with the supplementation of fodder, crop residues, and maize bran at 1–2 kg during milking. Furthermore, TALIRI-Tanga has a 0.5 ha experimental field that is used to plant and grow different types of forages that are distributed to smallholder dairy farmers in the Tanga region. The dairy farm also has tractors, fodder chaff cutters, harvesters, two water tanks for rainwater harvesting, feed baskets, and fodder grinders. The harvested fodder is used to make silage, which is fed to the lactating dairy cows during the hot season. Lactating dairy cows have free access to clean drinking water at the milking parlor and a commercial mineral salt supplement in a shaded shelter. The herd management system records the cow’s information, including the date of birth, calving date, body weight, parity, days in milk, lactation number, date of insemination or natural breeding, and health status. The most common tick-borne diseases regularly treated on the farm include *Theileriosis*, *Babesiosis*, *Anaplasmosis*, and internal parasitic worm infestation. Furthermore, dairy cows are also monitored on a daily basis, and any health problems or incidents are treated immediately by the veterinarian of TALIRI-Tanga. There is regular deworming and vaccination and the prevention of parasitic diseases through acaracide administration by dipping the animals and flushing the drugs, with a record of all drugs administered to the cows.

### 2.3. Animal Selection, Management, and Experimental Design

This study was conducted from 1 January 2022 to 28 February 2022 (hot season) and from 1 June up to 31 July 2022 (cool season) (Figure 1). TALIRI Tanga dairy farm was purposely selected, as it is located in the eastern lowlands, where the temperatures and humidity are high and the institute has facilities that are conducive for the research work. The farm keeps Holstein Friesian crosses with 50 (HF50) and 75% (HF75) Friesian gene levels. A total of 29 lactating Holstein Friesian crossbred dairy cows were selected based on the following criteria: the cow having two–three parities, being in the second to third month of lactation, freedom from lameness, and the absence of any other signs of health disorders. Sixten lactating Holstein Friesian crossbred dairy cows were selected among 71 lactating dairy cows in the hot season, whilst 13 were selected among 73 in the cool season. The majority of lactating dairy cows that were under milking during the study period were Sahiwal, Boran, and crosses of Ayrshire and Zebu. Table 1 shows the number of lactating Holstein Friesian crossbred dairy cows used in this study for each genotype, parity, and lactation month. The management practices for all experimental lactating Holstein Friesian crossbred dairy cows in the two seasons were similar. The lactating Holstein Friesian crossbred dairy cows were freely grazed with other cows from 08:00 to 12:00 h and 17:00 to −18:30 h in a farm with approximatively 15 ha of natural pasture and Napier grasses. All experimental lactating Holstein Friesian crossbred dairy cows were supplemented with 1 kg of concentrate in the morning and 1 kg in the evening, making the total amount of concentrate provided per day equal to 2 kg during the milking time. The ingredients and chemical composition of the concentrate diet given to the lactating Holstein Friesian crossbred dairy cows are shown in Table 2. The lactating Holstein Friesian crossbred dairy cows had access to drinking water at all times. The lactating Holstein Friesian crossbred dairy cows were weighed before the start of the experiment and at the end of the experiment.

### 2.4. Data Collection for the Milk Yield and Milk Composition Parameters

Daily milk yield records and physiological parameters were collected in the morning and evening during milking time (04:00–06:00 h and 14:30–17:00 h). Lactating Holstein Friesian crossbred dairy cows were milked through hand milking. The amount of milk produced by each cow was measured using calibrated milking cups immediately after milking in the morning and evening hours. For milk composition determination, a total of 50 mL of milk per cow was collected weekly on Monday morning and afternoon, put in the falcon tube, and then stored in the cool box. The milk samples were analyzed for protein, fat, lactose, and solids–not–fat on the next day using a Lactoscan machine (Lactoscan MCC-K3051(MILKOTRONIC LTD, Nova Zagora, Bulgaria)) [37]. The milk sample was placed at room temperature to yield the best results. Briefly, before starting the milk sample analyses, the milk analyzer was cleaned using warm water and 3% acidic solution (Lactoweekly cleaning solution) and again with warm water. After cleaning the Lactoscan machine, the milk sample was placed in the tube where the machine was taking 25 mL of the milk sample, and it displayed the milk parameter values after 1 min. The parameters analyzed using a Lactoscan machine include the milk sample temperature, fat, protein, lactose, solids–not–fat percentages, density, added water, electrical conductivity, pH, and salt.

### 2.5. Data Collection for Physiological Parameters

The data recorded for physiological traits included the core-body temperature (CBT), respiration rate (RR), rectal temperature (RT), and panting score (PS), which were collected every day between 04:00 h and 06:00 h and between 14:30 h and 17:00 h. The CBT was measured using a digital infrared thermometer at the site located 20 cm below the vertebral column of the animal. Respiration rates were collected in seconds and were taken from standing cows to make five flank movements [27,38,39]. The RR were obtained by counting twice the number of breaths in the flank region for a period of 15 s. The average values were multiplied by four to obtain the number of breaths per minute. The respiration rate was only recorded while the animal was standing (±ruminating). When the animal was performing other activities (walking, grazing, social, or grooming behavior), the counting was stopped. The determination of RR was accomplished before milking at the milking parlor. The RT was measured using a veterinary digital thermometer inserted at 3 cm in the rectum for approximately 1 min. At the same time, the lactating Holstein Friesian crossbred dairy cows were observed in the morning and afternoon for signs of open mouth panting to determine the PS on a scale of zero to four (Table 3), where 0 = normal breathing, 1 = slightly panting, 2 = moderate panting, 3 = strong panting, and 4 = severe panting [40].

### 2.6. Calculation of the Temperature–Humidity Index

Climate data were obtained from the Tanga meteorological station located 500 m away from TALIRI Tanga dairy cow’s farm. Table 4 shows the environmental conditions during the experimental periods. The data collected included the daily maximum and minimum temperatures (°C) and the relative humidity (%). The daily THI was calculated using the equation of NRC [41], which is considered the most appropriate for the equatorial climate of Tanzania [19]; THI = (1.8 × Tmax + 32) − [(0.55 − 0.0055 × Rhmin) × (1.8 × Tmax − 26.8)], where Tmax is the maximum daily dry–bulb temperature (°C) and Rhmin is the minimum daily relative humidity (%). Each THI was computed using a 4-day average of the daily maximum dry bulb temperature (Tmax) and minimum relative humidity (Rhmin) obtained from measures on the test day and 3 days prior to the test day, as recommended by Ekine-Dzivenu et al., 2020 [19]. This range helps to determine the prolonged effects of HS on physiological and milk production parameters recorded on a particular day [19]. In this study, only the maximum daily THI was used in the analyses because milk yield traits and physiological parameters are more sensitive to the extreme values of the maximum THI relative to the daily average THI [27,42,43]. Moreover, the use of the maximum daily THI has been recommended by Ravagnolo and Misztal [24], who indicated that combining daily temperature and humidity values from public weather stations to define the THI shows a superior goodness of fit than other combinations under the hot and humid conditions. The computed daily THI values were categorized into three groups as follows: low (THI = 76–78), moderate (THI = 79–81), and high (THI = 82–84). This classification was made according to Moore et al. [44]: (1) class A: THI ≥ 76 ≤ 78 (no HS condition); (2) class B: THI ≥ 79 ≤ 81 (moderate HS condition); and (3) THI ≥ 82 ≤ 84 (severe HS condition).

### 2.7. Statistical Analysis

Data analysis was performed using the SAS 9.2 statistical package (SAS, 2003). Records for all traits measured in this study on 29 lactating Holstein Friesian crossbred dairy cows in both the hot and cool seasons were analyzed together. The PROC MEANS procedure of SAS 9.2 was used to calculate the descriptive statistics for the milk yield, milk composition, and physiological traits as well as the body weight of the animals. The effects of the THI, genotype, parity, months of lactation, and their interactions were analyzed using the PROC MIXED procedure of SAS 9.2. The statistical model included fixed effects for the THI, the genotype, the parity, the months of lactation, and the interactions between the THI and the genotype, parity, and months of lactation. The dependent variables were the daily milk yield, milk composition (fat, protein, lactose and solids–not–fat), and physiological (CBT, RR, RT, and PS) parameters. The final mixed linear model for each parameter is as follows:Y_ijklmnpr_ = μ + THI_i_ + G_j_ + P_k_ + L_ℓ_ + (THI × G)_ij_ + (THI×P)_ik_ + (THI × L)_iℓ_
where Y_ijkℓmnpr_ is the phenotypic records for the milk yield or milk composition or physiological parameters; *µ* is the overall mean; THIi is the effect of the i^th^THI class; G_j_, P_k_, and L_ℓ_ are the effects of the genotype, parity, and months of lactation, respectively; (THI*G)_ij_ is the effect of interaction of the i^th^THI class and ^jth^G; (THI*P)_ik_ is the effect of the interaction of the i^th^THI class and k^th^ parity; and (THI*L)_iℓ_ is the effect of the interaction of the i^th^THI class and ℓ^th^ months of lactation. The effects of the season (hot and cool) and milking time (morning and afternoon) were confounded within the THI. Thus, they were removed from the model. For all models, the significance of the differences between the pairs of means was tested using the Tukey–Kramer test, and significance was declared at *p* ≤ 0.05. The PROC REG and PROC CORR procedures of SAS 9.2 were used to determine the linear and nonlinear relationships and Pearson correlation coefficients between the THI values and the milk yield, milk composition, and physiological parameters to better evaluate the relationship between these parameters and THI.

## 3. Results

### 3.1. Descriptive Statistic for the Milk Yield and Milk Composition Parameters of Lactating Holstein Friesian Crossbred Dairy Cows

The summarized statistics for the milk yield and milk composition traits measured for 29 lactating Holstein Friesian crossbred dairy cows during the hot and cool seasons are presented in Table 5. These statistics, including the number of observations, mean, minimum, maximum, and standard deviation, are presented according to the season, genotype, parity, lactation month, and milking time.


*Descriptive statistics for the physiological parameters of lactating Holstein Friesian crossbred dairy cows.*


The summarized statistics for the physiological traits measured for the 29 lactating Holstein Friesian crossbred dairy cows during the hot and cool seasons are presented in Table 6. These statistics, including the number of observations, mean, minimum, maximum, and standard deviation, are presented according to the season, genotype, parity, lactation month, and milking time.

### 3.2. Effect of THI, Genotype, Parity, and Months of Lactation on the Milk Yield and Milk Composition Parameters of Lactating Holstein Friesian Crossbred Dairy Cows

In this study, the THI, genotype, parity, and months of lactation had significant effect on the milk yield, milk fat, protein, lactose, and solids–not–fat percentage (*p* ˂ 0.05). The daily milk yield, fat, protein, lactose, and solids–not–fat percentages increased (*p* ˂ 0.05) slightly with increasing THI values from 76 to 78, remained fairly constant for the THI values of 79 to 81, and declined for the THI values of 82–84. In short, no decrease was observed at low THI values of 76 to 78, and there was a higher decrease at 82 to 84. In terms of the milk composition, the average fat, protein, lactose, and solids–not–fat percentages significantly declined as the THI increased. Regarding the genotype, the HF75 showed a higher milk yield (3.55 ± 0.04 L/day) compared to HF50 (3.03 ± 0.04 L/day). However, the HF50 showed a higher milk fat, protein, and lactose percentage compared with the HF75. In the present study, parity influenced the milk yield and composition parameters such that the daily milk yield declined (*p* < 0.05) from the second to third parity. Similarly, the milk fat, protein, lactose, and solids–not–fat percentages declined (*p* < 0.05) from the second to third parity. In short, lactating Holstein Friesian crossbred dairy cows in the second parity showed a higher milk yield and milk composition compared with the third parity lactating Holstein Friesian crossbred dairy cows. Similarly, the months of lactation also had a significant effect on the milk yield and composition parameters such that the milk yield declined (*p* < 0.05) from the second month (3.41 ± 0.03 L/day) to the third month of lactation (3.09 ± 0.03 L/day), whilst the milk fat, protein, lactose, and solids–not–fat percentages increased from the second to third months of lactation (Table 7).

### 3.3. Effect of THI, Genotype, Parity, and Months of Lactation on Physiological Parameters of Lactating Holstein Friesian Crossbred Dairy Cows

In this study, the THI, genotype, parity, and months of lactation month had a significant effect on the physiological parameters (*p* ˂ 0.05). There was a small increase in physiological parameters at low THI values of 76 to 78, whilst a higher increase in these parameters was observed at high THI values of 82 to 84. Generally, when the THI ranged from 76 to 78 and 79 to 81, lactating Holstein Friesian crossbred dairy cows had low to moderate HS, as indicated by all physiological parameters. Lactating Holstein Friesian crossbred dairy cows respired significantly faster (72.78 ± 0.29 breaths/min) and panted relatively more frequently (1.46 ± 0.01) when the THI values ranged from 82 to 84. The average CBT and RT of the lactating Holstein Friesian crossbred dairy cows followed a similar trend, with significantly higher RT and CBT values recorded when the THI was above 82–84.

Regarding the genotype, both HF50 and HF75 showed similar increases in all physiological parameters (Table 8). In the present study, parity also influenced physiological parameters such that these parameters declined (*p* < 0.05) slightly from the second to third parities. In short, lactating Holstein Friesian crossbred dairy cows in the third parity showed a small increase in CBT, RT, RR, and PS compared with the lactating Holstein Friesian crossbred dairy cows in the second parity. Generally, both lactating Holstein Friesian crossbred dairy cows in the second month and third months of lactation showed similar increases in the physiological parameters. The months of lactation also influenced physiological parameters such that lactating Holstein Friesian crossbred dairy cows in the third month of lactation showed a slightly higher increase in CBT, RT, and RR compared with lactating Holstein Friesian crossbred dairy cows in the second month of lactation (*p* < 0.05) (Table 8).

### 3.4. Effect of the Interaction of the Genotype and THI on the Milk Yield, Milk Composition, and Physiological Parameters of Lactating Holstein Friesian Crossbred Dairy Cows

The effect of the genotype and THI interactions were significant for the milk yield and milk composition parameters (Table 9). The lactating Holstein Friesian crossbred dairy cows with an HF75 gene level showed a decline in the milk yield from 3.53 ± 0.10 to 3.45 ± 0.05 L/day when the THI increased from low (76–78) to high (82–84) THI values. In contrast, the lactating Holstein Friesian crossbred dairy cows with an HF50 gene level showed an increase in the milk yield from 2.68 ± 0.08 L/day to 3.58 ± 0.04 L/day as the THI increased from low to high THI values. For both HF50 and HF75, the milk fat content slightly increased when the THI increased from low (76–78) to high (82–84) THI values, but the increase in the milk fat content was larger in HF75 than in HF50. Moreover, the protein, lactose, and solids–not–fat percentages increased (*p* < 0.05) for both HF50 and HF75 lactating dairy cows when the THI values ranged from low to moderate THI and then slightly declined at the same rate when the THI changed from moderate to high THI values.

Regarding physiological parameters, both HF50 and HF75 lactating dairy cows showed a slight increase (*p* < 0.05) in all physiological parameters as the THI increased from low to high THI values. The HF50 lactating dairy cows respired and panted at significantly higher rates at moderate (62.97 ± 0.41 breaths/min) and high THI values (72.72 ± 0.40 breaths/min) compared with HF75, which showed 62.29 ± 0.43 breaths/min at moderate THI and 72.52 ± 0.41 breaths/min at high THI values. Overall, HF50 and HF75 indicated similar HS patterns for physiological parameters as the THI values increased from low to high THI thresholds (Table 9).

### 3.5. Effect of Parity and THI on the Milk Yield, Milk Composition, and Physiological Parameters of Lactating Holstein Friesian Crossbred Dairy Cows

The effect of parity and THI interactions on the milk yield, milk composition, and physiological parameters were significant (Table 10). Lactating Holstein Friesian crossbred dairy cows in the second parity showed a higher milk yield per day (3.52 ± 0.05 L/day and 3.68 ± 0.04 L/day) compared with lactating Holstein Friesian crossbred dairy cows in the third parity (3.25 ± 0.04 L/day and 3.60 ± 0.05 L/day) at the THI values of 79–81 and 82–84, respectively. Generally, lactating Holstein Friesian crossbred dairy cows in the third parity showed a low milk yield per day compared to those in the second parity. Furthermore, lactating Holstein Friesian crossbred dairy cows in the second parity had a higher milk fat content (2.75 ± 0.06% and 2.67 ± 0.05%) compared with lactating Holstein Friesian crossbred dairy cows in the third parity (2.73 ± 0.05% and 2.64 ± 0.06%) at the THI values of 76–78 and 79–81, respectively. Nevertheless, lactating Holstein Friesian crossbred dairy cows in the second and third parities had similar contents of milk protein, milk lactose, and milk solids–not–fat, and similar reduction patterns were observed when the THI increased from moderate (79–81) to high (82–84) THI values.

Regarding physiological parameters, lactating Holstein Friesian crossbred dairy cows in the third parity respired significantly faster (75.50 ± 0.41 breaths/min) and panted relatively more frequently (1.50 ± 0.02) than the lactating Holstein Friesian crossbred dairy cows in the second parity, which showed an RR of 75.52 ± 0.30 breaths/min and a PS of 1.50 ± 0.01 at THI thresholds of 82–84 (Table 10).

### 3.6. Effect of Months of Lactation and THI on Milk Yield, Milk Composition, and Physiological Parameters of Lactating Holstein Friesian Crossbred Dairy Cows

Lactating Holstein Friesian crossbred dairy cows in the second and third months of lactation showed a slight increase (*p* < 0.05) in the milk yield as the THI values increased from low to high THI values. Lactating Holstein Friesian crossbred dairy cows in the second month of lactation also showed a high milk yield per day (3.38 ± 0.04 L/day and 3.87 ± 0.05 L/day) compared to the lactating Holstein Friesian crossbred dairy cows in the third month of lactation (3.25 ± 0.07 L/day and 3.18 ± 0.06 L/day), respectively, at moderate to high THI thresholds. However, lactating Holstein Friesian crossbred dairy cows in the third month of lactation showed a higher milk fat content (2.66 ± 0.15% and 2.79 ± 0.10%) compared to the lactating Holstein Friesian crossbred dairy cows in the second month of lactation (2.62 ± 0.08% and 2.78 ± 0.05%) at THI values of 76–78 and 79–81, respectively. In both the second and third months of lactation, the Holstein Friesian crossbred dairy cows showed similar patterns of an increase (*p* < 0.05) in protein, lactose, and solid–not–fat percentages from low to moderate THI values, with similar reduction patterns when the THI thresholds increased from moderate to high THI values (Table 11).

Regarding physiological parameters, lactating Holstein Friesian crossbred dairy cows in both the second and third months of lactation showed similar patterns of an increase in all physiological parameters. Lactating Holstein Friesian crossbred dairy cows respired more frequently, with 71.47 ± 0.42 breaths/min and 73.66 ± 0.41 breaths/min in the second and third months of lactation at THI values of 82–84, respectively. There was also a slight panting score of 1.44 ± 0.02 in the second month and 1.48 ± 0.02 in the third month of lactation at THI values of 82–84 (Table 11).

### 3.7. Pearson Correlation Coefficients between THI and Milk Yield, Milk Composition, and Physiological Parameters of Lactating Holstein Friesian Crossbred Dairy Cows

In this study, milk yield (r = 0.24, *p* < 0.0001) and fat % (r = 0.15, *p* ≤ 0.05) were positively correlated with THI, an indication that THI had less of an influence on the milk yield and milk fat percentage. However, the milk protein % (r = −0.15, *p* ≤ 0.05), milk lactose (r = −0.13, *p* ≤ 0.05), and milk solids–not–fat percentages (r = −0.14, *p* ≤ 0.05) were negatively correlated with the THI. On the other hand, the milk composition parameters showed a highly positive correlation between themselves (Table 12). Furthermore, a highly significant positive correlation between THI and all the physiological parameters was observed. The CBT (r = 0.67, *p* < 0.0001), RT (r = 0.63, *p* < 0.0001), and RR (r = 0.63, *p* < 0.0001) were highly positively correlated with THI, whilst the PS (r = 0.16, *p* < 0.0001) showed a low positive correlation with THI (Table 13).

### 3.8. Relationship between THI, Milk Yield, and Composition Parameters of Lactating Holstein Friesian Crossbred Dairy Cows

In this study, the milk yield and fat percentage showed a low positive association with THI (*p* < 0.05) (Figure 2 and Figure 3). However, the protein, lactose, and solids–not–fat percentage showed a low negative correlation with THI (*p* < 0.05) (Figure 4, Figure 5 and Figure 6).

### 3.9. Relationship between THI and Physiological Parameters

In the current study, the CBT (R^2^ = 0.93), RT (R^2^ = 0.64), RR (R^2^ = 0.90), and PS (R^2^ = 0.41) showed a moderate to strong positive association with THI (*p* < 0.05) (Figure 7, Figure 8, Figure 9 and Figure 10).

## 4. Discussion

Most dairy cows kept in Tanzania are crosses of European dairy breeds (mostly Friesian and Ayrshire) with Tanzania Shorthorn zebu/Boran. Dairy cows kept along the coast might experience heat stress since the coastal lowland has high temperatures and humidities, with THI values reaching above 77.29 in the hot season [4]. This study assessed the effects of THI on the daily milk yield, milk composition, and physiological parameters of lactating Holstein Friesian crossbred dairy cows kept at TALIRI, Tanga, which is located at an altitude of 6 m above and is 6 km from the Indian Ocean.

In this study, the results show that the mean ambient temperature and THI values in the hot season (January–February) were higher than those in the cool season (June–July). On the other hand, the mean relative humidity in the cool season was higher than that in the hot season. Generally, the observed mean THI values were high in such a way that they can cause HS to the animals. Similar findings were reported by Lim et al. [45] in their study about HS effects on the milk production parameters of Holstein and Jersey lactating dairy cows in South Korea. Some studies reported that the dairy cow’s thermoneutral zone ranges between 5 °C and 25 °C but can fall to the range from 0.5 °C to 20 °C and 60% to 80%RH, although this varies depending on the production status, feed type, acclimatization level, and climatic conditions [18,45]. Habimana et al. [18] reported that when the THI exceeds 72, dairy cows begin to experience HS. In the present study, a decline in milk yield and composition parameters was observed when the THI values ranged from 77 to 84. In Rwanda, Niyonzima et al. [30] reported THI values ranging from 63.3 to 84.6 with an average of 75.8 THI as the HS threshold for the milk yield decline. This is in agreement with the THI thresholds obtained in this study. In Florida (USA), Fabris et al. [46] reported THI values of above 68 as HS thresholds for lactating dairy cows. In India, Velayudhan et al. [47] reported THI values of 72–75 as the most favorable for lactating dairy cows in the tropical region of Bengaluru because of the maximum milk production obtained. In the smallholder farms of Tanzania, Ekine-Dzivenu et al. [19] reported a THI of 76 as the HS threshold. The seasonal changes in THI thresholds and the associated variations in the milk yield, milk composition, and physiological parameters observed in this study could be associated with variations in weather conditions over the months. These climatic variations lead to alterations in the quality and quantity of the diet provided to dairy cattle [19].

### 4.1. Heat Stress Effects on the Milk Yield and Composition Parameters of Lactating Holstein Friesian Crossbred Dairy Cows

The results in the present study indicate that the milk yield and fat percentage increased slightly, whereas the milk protein, lactose, and solids–not–fat percentages decreased, when the THI values increased from 76 to 84. One possible reason for this could be that a slight increase in the milk yield as the THI increased implies that there was very little influence of THI values above the threshold on milk production [30]. These results are inconsistent with the findings by Sungkhapreecha et al. [48] in Thailand, who reported that the milk yield declines when the THI reach 76. In Australia, Talukder et al. [43] reported that the daily milk yield increased as the THI increased up to a THI value of 65 and remained fairly constant until 85, decreasing afterwards. This is in agreement with the findings of this study. In Brazil, Stumpf et al. [49] indicated that an increase in THI thresholds generally causes a decline in milk production parameters, which is in contrast with the findings of this study. Under the Mediterranean climatic conditions of Tunisia, Bouraoui et al. [20] reported a decline in the milk fat percentage of 0.34% and a 21% decline in the milk yield when the THI values increased from 68 to 78 for lactating Holstein Friesian dairy cows. In Rwanda, Niyonzima et al. [30] reported a decrease in the milk yield for lactating Holstein crossbred dairy cows when the THI thresholds were above 76, which is not in agreement with the findings of this study. The decline in milk yield traits during HS could be the result of a reduction in the feed intake and decreased nutrient uptake by the portal drained viscera of dairy cows [19,20,27]. In this study, HS reduced the milk yield and milk composition parameters when the THI increased from moderate to high THI values for both HF50 and HF75 lactating dairy cows. There was a higher decrease in milk composition parameters for HF75 than for HF50 lactating dairy cows when the THI values increased from moderate to high THI thresholds, an indication that the later are relatively more heat-tolerant than the HF75. In Brazil, Alfonzo et al. [50] reported that Girolando-Holstein Friesian cows with a 75% gene level (GH75) were less tolerant to heat compared with GH50 cows, which is in agreement with the findings of this study.

In Germany, Lambertz et al. [7] reported a milk fat and protein decline as the THI increased from 60. Garcia et al. [51] indicated that under HS conditions, the higher fat content in dairy cows’ milk is caused by an increase in free fatty acids during the negative energy balance, whilst a decline in milk protein is due to a lower synthesis of casein formation enzymes in the mammary gland. In the study by Lambertz et al. [7] across Holstein Friesian dairy genotypes, the milk protein, lactose, and solids-not-fat declined when the THI changed from moderate to high THI values. This is in agreement with the findings of this study. The findings of this study also partially agree with those reported by Corazzin et al. [15], who observed an increase in fat and protein percentages but no increase in the lactose percentage. This shows that HS decreases the protein content of milk without affecting the fat percentages. Also, the results in the present study concur with the results reported by Bouraoui et al. [20], who found that HS reduced milk fat and protein percentages when the season changed from spring to summer. Generally, HS effects on milk fat and protein percentages are largely non-consistent, as reported by Corazzin et al. [15]. Rhoads et al. [52] reported a reduction in the milk protein percentage of 0.13% during HS conditions. Moreover, individual animal differences and trait responses to HS are expected owing to animal-related factors like the breed and physiological responses such as the age, production status, feed intake, and animal behaviors [33]. The decline in the milk protein percentage detected in this study is in agreement with the results reported by Bouraoui et al. [20] and Rhoads et al. [52]. The milk protein concentration is determined by the energy absorption or the energy content of the diet, and its noneffective supply causes a decline in milk protein percentages [7]. It is well recognized that HS decreases feed intake [17,28], but the feed intake in animals grazing on pasture declines owing to the feed shortage during hot weather conditions [30]. The lactating Holstein Friesian crossbred dairy cows in this study were freely grazing on pasture, and the dry matter intake from the pasture was not recorded. Therefore, the lower milk yield observed in this study was a result of the reduced feed intake from the pasture combined with the physiological and metabolic effects of HS [7,30].

On the other hand, the genotype, parity, and months of lactation also influenced the milk yield and composition parameters. Corazzin et al. [15] reported that the higher percentage of protein and fat observed in the milk of heat-stressed dairy cattle could be the result of a decline in milk production and a subsequent increased concentration of protein and fat in addition to possibly greater non-protein nitrogen contents in the milk produced by dairy cows under HS conditions. Furthermore, the results of the current study show that parity influenced the milk yield and composition parameters such that the milk yield decreased from the second parity to the third parity. Similarly, the milk composition parameters declined from the second to third parities. This could be due the differences in feeding rates, as primiparous dairy cows eat more slowly than multiparous ones during the peak lactation [53]. Additionally, Moore at al. [44] reported that high-producing lactating dairy cows such as multiparous lactating dairy cows show major heat sensitivity owing to increased intrinsic metabolic heat production compared to young lactating dairy cows in the second parity. In the study by Sabek et al. [54], theparity and lactation month had negative effect on the milk yield, fat, protein, lactose, and solids–not–fat percentages, which is in agreement with the findings of this study.

Regarding the genotype and THI interaction, the findings showed different trends for the milk yield and composition of HF50 and HF75 lactating dairy cows when the THI values increased. There was a marked difference between HF50 and HF75 in the milk yield at THI values of 76 to 78. The milk yield declined and increased in HF75 and HF50 lactating dairy cows, respectively, when the THI values increased from moderate to high. Regarding milk composition parameters, HF50 showed higher contents of fat, protein, lactose, and solids–not–fat percentages than HF75. Nevertheless, the fat % declined when the THI changed from moderate to high THI values in both HF50 and HF75, whilst no reduction was observed for protein %. No large difference was observed for the lactose percentage in both HF50 and HF75 when the THI changed from low to high THI values. A large decline in solids–not–fat was observed in HF50 compared to that in HF75 when the THI changed from moderate to high THI values. These findings are in agreement with those reported by Stumpf et al. [49], who observed a greater decrease in milk production for HF75 than for HF50 when the THI values increased from 79.2 to 80.32. The magnitude of the milk yield decline and the alteration of milk composition parameters including fat, protein, lactose, and solids–not–fat percentages as a result of HS are influenced by various mechanisms at different lactation stages, and the mammary gland of lactating dairy cows respond differently to HS [27].

Parity and THI interaction influenced the milk yield and composition such that the milk yield decreased when the THI increased for the cows in both the second and third parities. These findings concur with those reported by Song et al. [55]. Lactating Holstein Friesian crossbred dairy cows in the second parity showed a higher milk yield and composition compared to those in the third parity, which indicates that multiparous lactating dairy cows in the second parity are highly tolerant compared to multiparous dairy cows in the third parity. In the study by Ouellet et al. [56], lactating dairy cows in all parities were stressed by HS and responded by showing a significant decline in milk protein percentages; this is inconsistent with the results of this study. Song et al. [55] indicated that primiparous lactating dairy cows are lighter than multiparous ones; thus, the ratio of the surface area to the volume is slightly higher, and this predisposes them to heat loss. Moreover, limited studies have explored the relationship between the lactation stages of dairy cows and THI values [57]. In this study, it was observed that lactating Holstein Friesian crossbred dairy cows in the third month of lactation showed a significant decline in the milk yield compared to those in the second month of lactation. However, the reverse was observed for fat, protein, lactose, and solids–not–fat percentages, whereby significant effects of HS were observed in terms of reduced milk composition traits for the cows in the second month of lactation compared to those in the third month of lactation.

Lactating Holstein Friesian crossbred dairy cows in the third month of lactation were highly affected by HS compared to those in the second month of lactation, as they showed a higher decline in the milk yield when the THI increased from low to high THI values. On the other hand, lactating Holstein Friesian crossbred dairy cows in the second month of lactation showed a higher decline in milk protein, lactose, and solids–not fat percentages compared to those in the third month of lactation as the THI increased from low to high THI values. This is in agreement with the findings of Lambertz et al. [7], who reported that the major effects of HS on the milk yield, fat, and protein percentages are identified in later lactation. Furthermore, the findings of the current study are supported by results from other studies, which reported that lactating dairy cows in the early stages of lactation are highly affected by HS in terms of productivity [42,57]. In Australia, Osei-Amponsah et al. [27] observed a significant effect of the stage of lactation on the daily milk yield, fat, and protein percentages, which is in agreement with the findings of this study. In a pasture–based system, Osei-Amponsah et al. [27] reported an increase in the milk fat % and protein % by 3% and 2%, respectively, when the THI changed from low to high THI values. The reduction in milk fat, protein, lactose, and solids–not–fat percentages is mainly influenced by the adverse effects of hot weather conditions on the synthesis of these milk constituents in the dairy cow mammary gland [42]. There are substantial milk yield losses induced by HS at any stage of lactation of dairy cows. Thus, the cooling of lactating dairy cows when the THI thresholds range between 77 and 84 is necessary through the use of trees in the farms, shading, the provision of drinking water, and the supplementation of concentrate during milking, among others, to minimize the decline in the milk yield observed in the afternoon and hot season. However, the different cooling approaches should be done with consideration of the production cost of the cooling technologies applied [16]. Under warm and humid conditions, dairy farmers could improve the milk yield and avoid fluctuations in the milk composition in different seasons through nutritional supplementation and the manipulation of feeding practices. Furthermore, the provision of fans, sprinklers, shade, barns, and trees, which enhance passive ventilation, could improve body heat loss, increase the dry matter intake of cows, and, hence, improve the dairy cattle milk composition [18,28,58].

### 4.2. Heat Stress Effects on the Physiological Parameters of Lactating Holstein Friesian Crossbred Dairy Cows

Studies have shown that an RR greater than 60 breaths/min indicates HS when dairy cow use evapotranspiration as the key mechanism for losing body temperature [51]. In the present study, when THI values increased from 76 to 84, the lactating Holstein Friesian crossbred dairy cows responded by increasing the RR. Also, when the THI values increased from 76 to 84, there was an increase of 0.4 °C for CBT and RT, and there were 11 breaths per min. Similar patterns of responses to HS were observed in the study conducted in Tunisia by Djelailia et al. [59]. In their study, HS altered RT, RR, and HR such that a daily increase of 1.2 °C was observed when the THI values increased from 55 to 78, while the HR and RR increased by 3 beats per min and 35 breaths per min, respectively. The physiological responses to HS observed in this study are an adaptive mechanism initiated by lactating Holstein Friesian crossbred dairy cows in an attempt to restore their thermal balance [59]. The CBT of lactating dairy cows varies many times, as it is a crucial tool for regulating body temperature and relies on the peripheral blood flow [45]. When a lactating dairy cow wants to reduce its body temperature, its body heat is transported from the core of the body to the skin by blood; thus, the blood flow to the skin will rise, thereby increasing the skin temperature [45].

The RT is considered to be a good indicator of deep CBT, although there are significant changes among various parts of the core body at different scales of the day [45]. In this study, significantly higher CBT and RT values were recorded when the THI values were above 82–84. These results are in agreement with those reported by Garcia et al. [51] in Brazil, who found a significant effect of THI on RT and RR. Also, the findings in this study are in agreement with those of Zhou et al. [60], who reported that the RT starts to rise when the ambient temperature reaches above 20 °C. In this study, the average RT increased by 0.4 °C when the THI thresholds ranged from 76 to 84. The RT was higher at 38.4 °C for HF50 compared to for HF75 at extreme THI values of 82–84, suggesting that HF50 cows were slightly less heat-tolerant than HF75 cows. However, other physiological parameters indicated that they were better tolerant to heat stress compared to HF75 cows. All lactating Holstein Friesian crossbred dairy cows showed mild to moderate HS for all physiological parameters when the THI ranged from 76 to 78 and 79 to 81. For instance, the high PS value observed at higher THI values and afternoon hours was comparable with first-phase panting and is the point at which HS mitigation should be considered [40]. During HS, lactating dairy cows increase RR and PS, which increases body fluid loss and affects dehydration and blood homeostasis [45].

In this study, the lactating Holstein Friesian crossbred dairy cows respired significantly faster and panted relatively more frequently at higher THI values. These results differ from those obtained by Djelailia et al. [59], who reported 63 breaths/min at a THI of 80 for Holstein Friesian dairy cows reared in the Mediterranean climate of Tunisia. The increased rate of RR and PS is an indication that these animals are losing heat as an attempt to maintain homeothermy [27]. Respiration rates increase when the ambient temperature surpasses the dairy cow thermoneutral zone, which typically ranges from −5 °C to 25 °C, and declines again below this thermoneutral zone [61]. The increase in RR in lactating dairy cow is used to disperse around 30% of body heat by respiratory vaporization. This respiratory vaporization and convection dissipation of body heat help a lactating dairy cow to maintain its thermal balance [29]. The RR has been shown to be the first physiological response to an increased ambient temperature for lactating Holstein Friesian dairy cows in the late stages of lactation reared in the Netherlands [16,60].

In the present study, lactating Holstein Friesian crossbred dairy cows in the third month of lactation indicated a slight increase in CBT, RT, RR, and PS compared to those in the second month of lactation, an indication that this group of animals experiences higher HS effects than their counterparts. These findings partially concur with those reported by Yan et al. [57], who found that the stage of lactation significantly influences the thresholds for the surface temperature maximum, but with a less significant effect on the surface temperature average. Yan et al. [57] also reported that lactating dairy cows in the third month of lactation are more susceptible to increases in HS conditions than those in the second month and first month of lactation, and this is in agreement with the findings of the current study. The findings of this study are in partial agreement with those reported by Osei-Amponsah et al. [27] for lactating Holstein dairy cows grazing during the Australian summer in Melbourne. In their study, the lactation stage had no significant effect on the RR, PS, and ST but affected (*p* ≤ 0.05) the average daily milk yield and milk solids. Yan et al. [16] indicated that some lactating dairy cows may transition from the previous stage of lactation to the next stage during the research period. This results in a failure to detect the potential effect of the stage of lactation. Moreover, parity influences the physiological responses of lactating dairy cows to HS [16]. In this study, lactating Holstein Friesian crossbred dairy cows in the third parity showed higher patterns of responses to HS with significant increases in CBT, RT, RR, and PS compared to those in the second parity. These findings concur with those reported by [16]. In their study, they analyzed the effects of parity on RT and RR and found that lactating dairy cows in the third parity had a higher RR compared with the lactating dairy cows in the first and second parities. Our findings also concur with those reported by Djelailia et al. [59], indicating that parity affects RT and RR.

### 4.3. Relationship between THI, Milk Yield, and Composition Parameters of Lactating Holstein Friesian Crossbred Dairy Cows

During HS conditions, lactating dairy cows exhibit several behavioral and physiological conditions that have negative effects on the milk yield and composition parameters [56]. In this study, the THI showed a low positive correlation with the milk yield and fat percentage but was negatively correlated with the protein, lactose, and solids–not–fat percentages. Bokharaeian et al. [62] reported a significant negative relationship between the THI and the milk yield, fat, protein, lactose, and solids–not–fat percentage, which is in partial agreement with the findings of this study. The findings in this study are also in contrast with those reported by Bouraoui et al. [20], who observed a negative relationship between the milk yield and THI. These findings partially concur with those reported by Bernabucci et al. [14], who fitted a linear model on a large dataset of lactating Italian Holstein dairy cow milk yield records and observed a significant negative relationship between extreme THI thresholds and milk production parameters. These findings also partially concur with those reported by Lim et al. [61], with a negative relationship between the milk yield and THI for lactating Holstein dairy cows and a positive association between the latter and the milk yield for Jersey dairy cows. A possible reason could be that a low positive correlation between the THI and the milk yield and the fat percentage observed in this study implies that the milk yield and fat tend to increase slightly as the THI increases [62].

These results suggest that milk composition parameters are more sensitive than the milk yield to the effects of HS [56]. In the Mediterranean climate of Italy, Bertocchi et al. [42] reported a strong positive correlation between THI and fat % (r = 0.98) and protein % (r = 0.99) for a retrospective study on lactating Holstein dairy cows for data collected between 2003 and 2009. However, these findings are partially consistent with the results of this study. Furthermore, the findings of this study partially concur with those reported in China by Yan et al. [16], who observed a positive correlation between the milk yield and THI values. In their study, they observed a positive correlation between the milk yield and THI and a positive association between the THI and milk fat %, milk protein %, and milk lactose.

### 4.4. Relationship between THI and Physiological Parameters of Lactating Holstein Friesian Crossbred Dairy Cows

In this study, THI showed moderate to strong positive correlation with all the physiological parameters, except the PS, which showed a low positive correlation with the THI. Similar findings were observed by Osei-Amponsah et al. [27], who reported that all physiological parameters measured in their study were positively correlated with THI. Osei-Amponsah et al. [27] reported a moderate positive correlation between THI and RR, PS, and CBT, which is in agreement with the findings of the present study. Djelailia et al. [59] also reported a positive correlation between RR, HR, and RT, an indication that these parameters are indicators of thermal stress and can be used to investigate the adverse effects of HS on lactating dairy cows. These findings also concur with those reported by Lim et al. [45] in South Korea, who found that THI had a strong correlation (r = 0.99) with the rumen surface temperature compared to RT, RR, and udder surface temperature in the lactating Holstein dairy cows and the RR (r = 0.97) compared with the RT, RST, and UST in the lactating Jersey dairy cows. Similar findings were also reported in the study by [63]. Djelailia et al. [59] reported that the positive correlations between THI and physiological parameters indicate the sensitivity of those parameters as indicators of responsiveness to the environment. There have been limited studies concerning the relationship between THI or other climatic variables and physiological parameters of lactating dairy cows in sub-Saharan countries, making it difficult to compare the results of this study.

## 5. Conclusions

The results of this study show that the daily milk yield and the values of the tested milk composition parameters declined while the physiological parameters (CBT, RT, RR, and PS) increased at THI thresholds ranging between 79 and 84. The results revealed that the milk yield and milk composition parameters (fat, protein, lactose, and SNF percentages) increased slightly with rising THI values for the THI thresholds of 76–78 and then significantly decreased when the THI exceeded 82–84. The decline in the daily milk yield and milk composition and the increase in physiological parameters were lower in the lactating HF50 dairy crosses than in the HF75, implying that the HF50 are more heat-tolerant compared to HF75 dairy cow crosses. Thus, the HF50 dairy crosses are better suited to the warm and humid conditions of Tanga region, Tanzania. Similarly, the lactating Holstein Friesian crossbred dairy cows in the second parity showed better tolerance to HS than those in the third parity. Moreover, lactating Holstein Friesian crossbred dairy cows in the second month of lactation were highly tolerant to HS than those in the third month of lactation. However, the study has demonstrated that both HF50 and HF75 lactating Holstein Friesian crossbreds reared in the eastern coastal lowlands of Tanzania experience HS, as indicated by a reduction in the milk yield and milk composition as well as an increase in the CBT, RT, RR, and PS when the THI values ranged from 82 to 84. There were moderate to strong positive correlations between THI and physiological parameters, but there were very low positive correlations between THI and the milk yield and fat percentage, whilst protein, lactose, and solid–not–fat percentages were negatively correlated with THI. It is recommended that mitigation strategies such as providing shade, cooling technologies, planting trees in the pasture farm, providing clean water and concentrate feeds, and the genetic development of heat-tolerant breeds need to be adopted and promoted in lowland warm and humid areas in Tanzania to support sustainable dairy cow farming under changing climatic conditions. Further studies with mathematical modeling describing the daily patterns and thresholds for THI can be useful in mitigating HS and providing alternative mitigation and production strategies.

## Figures and Tables

**Figure 1 animals-14-01914-f001:**
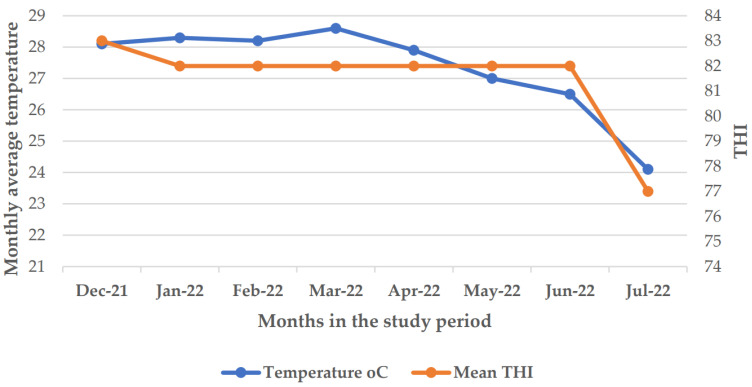
Average temperature (°C) and temperature—humidity index variation across the months of the study period.

**Figure 2 animals-14-01914-f002:**
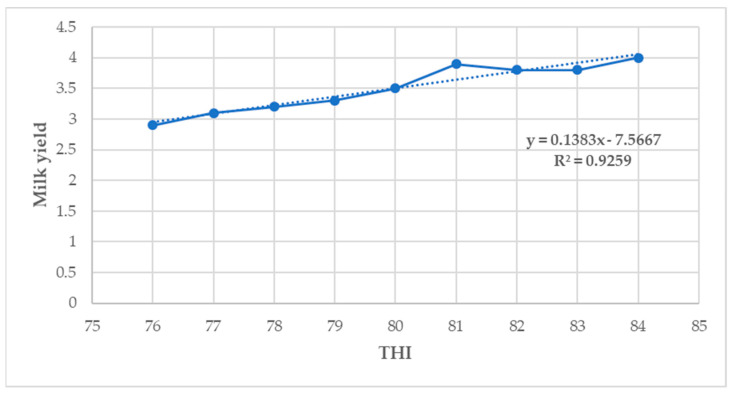
Relationship between THI and milk yield (*p* < 0.05).

**Figure 3 animals-14-01914-f003:**
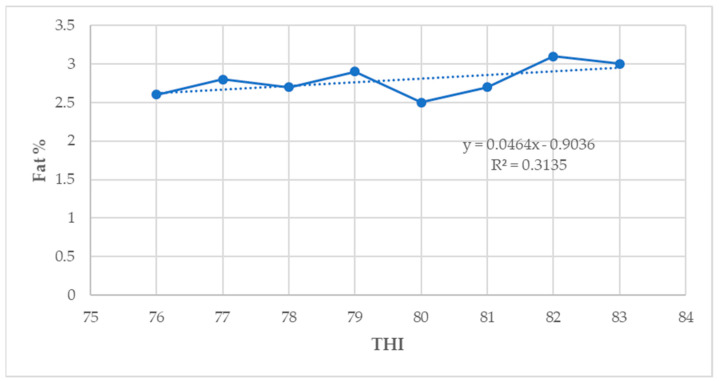
Relationship between THI and fat percentage (*p* < 0.05).

**Figure 4 animals-14-01914-f004:**
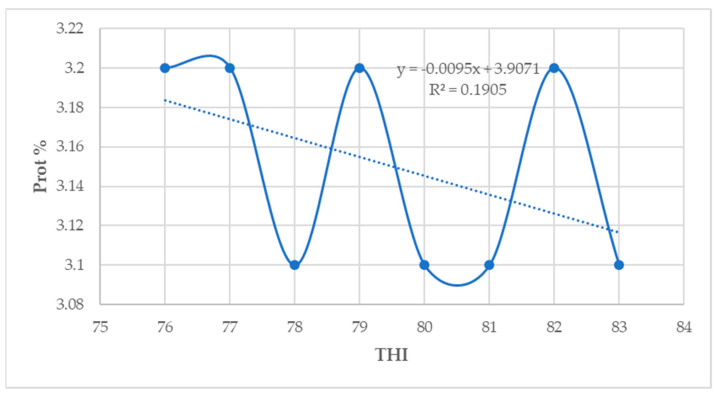
Relationship between THI and protein percentage (*p* < 0.05).

**Figure 5 animals-14-01914-f005:**
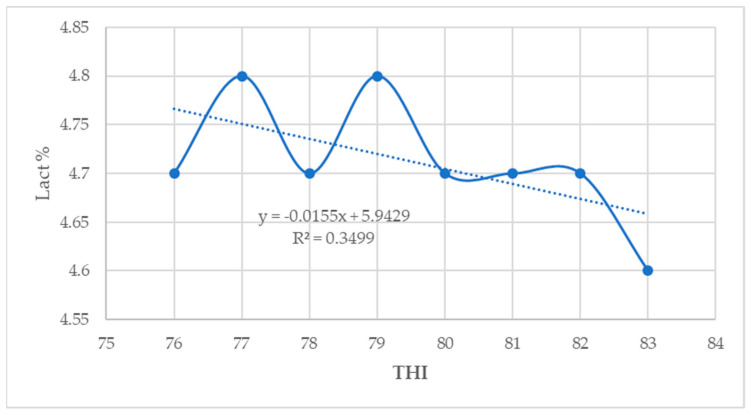
Relationship between THI and lactose percentage (*p* < 0.05).

**Figure 6 animals-14-01914-f006:**
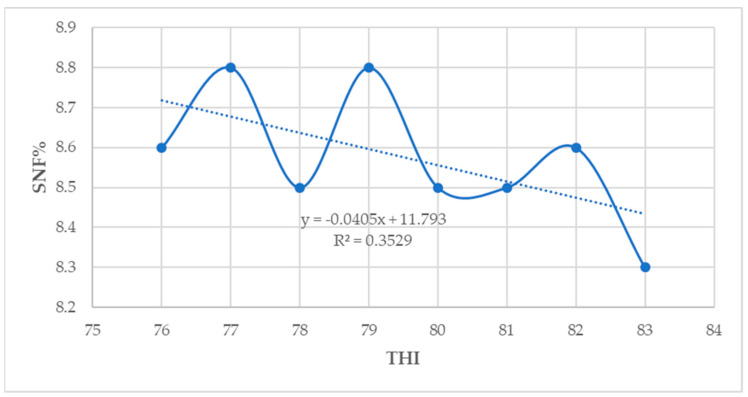
Relationship between THI and solids–not–fat percentage (*p* < 0.05).

**Figure 7 animals-14-01914-f007:**
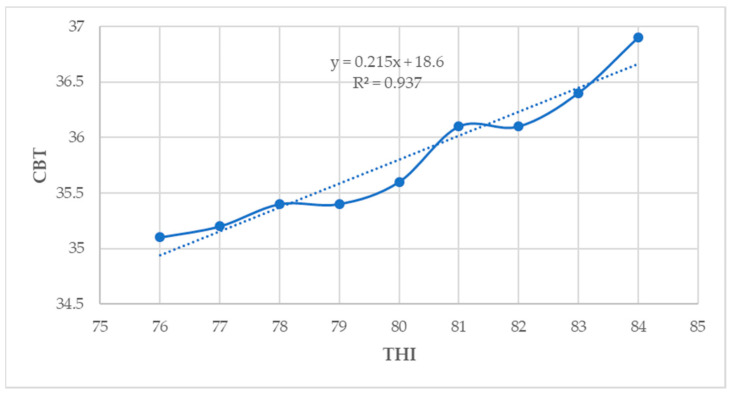
Relationship between THI and CBT (*p* < 0.05).

**Figure 8 animals-14-01914-f008:**
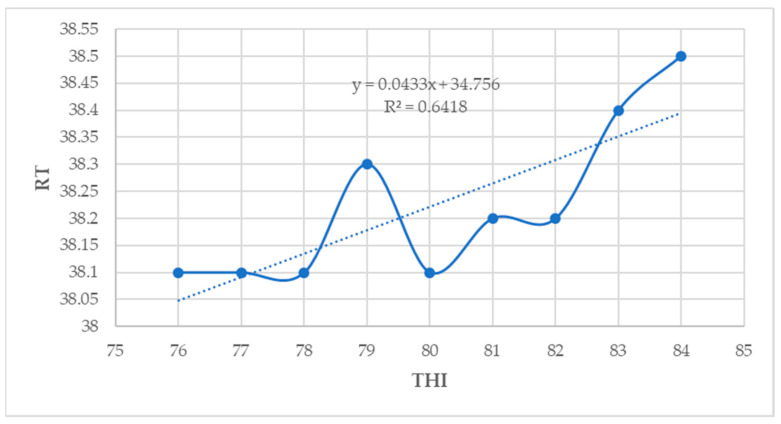
Relationship between THI and RT (*p* < 0.05).

**Figure 9 animals-14-01914-f009:**
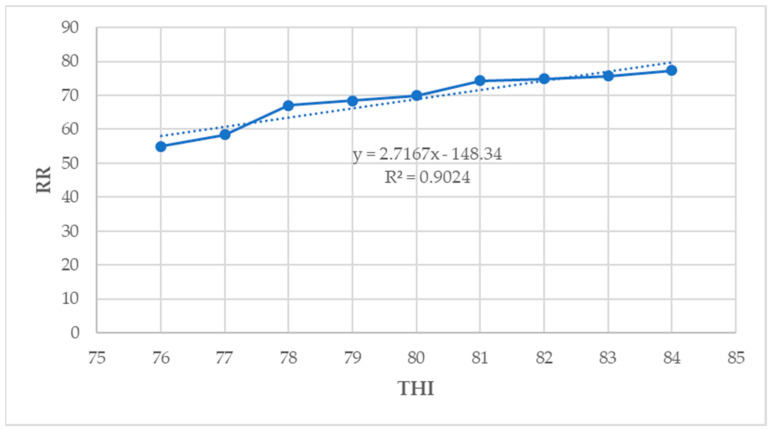
Relationship between THI and RR (*p* < 0.05).

**Figure 10 animals-14-01914-f010:**
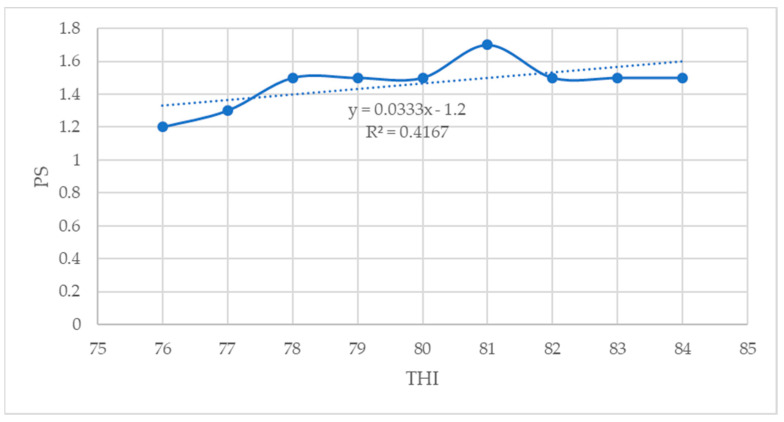
Relationship between THI and PS (*p* < 0.05).

**Table 1 animals-14-01914-t001:** Number of lactating Holstein Friesian crossbred dairy cows used in the study during the hot and cool seasons.

Item	Hot Season	Cool Season
Number of lactating Holstein Friesian crossbreds	16	13
HF50	10	8
HF75	6	5
Second parity	10	6
Third parity	6	7
Second month of lactation	11	10
Third month of lactation	5	3

Note: HF50 = Lactating Holstein Friesian crosses with a 50% gene level; HF75 = Lactating Holstein Friesian crosses with a 75% gene level.

**Table 2 animals-14-01914-t002:** Feed ingredients and chemical composition of the concentrate diet.

Item	Amount (g/kg DM)
Ingredients	-
Maize bran	750
Sunflower seed cake	195
Lucerne meal	25
Limestone	10
Kitchen salt	5
Josera/mineral mixture	15
Chemical composition of the mixed diet	-
Crude protein	16.30
Crude fiber	10.07
Crude fat	27.8
Metabolizable energy (MJ/kg DM)	10.6
Ash	3.5

**Table 3 animals-14-01914-t003:** Scale used for the respiratory rate and panting score.

Panting Score (PS)	Breathing Condition	Respiration Rate (RR) (breaths/min)
0	Normal panting—normal (difficult to see chest movement).	Respiration rate less than/equal to ≤40 breaths/min
1	Slight panting—mouth closed; no drool or foam; easy to see chest movement.	Respiration rate of 40–70 breaths/min
2	Fast panting-drool or foam present; no open mouth panting.	Respiration rate of 70–120 breaths/min
2.5	Like panting score 2 but with occasional open mouth; tongue not extended.	Respiration rate of 70–120 breaths/min.
3	Open mouth with some drooling; neck extended and head usually up.	Respiration rate of 120–160 breaths/min.
3.5	Like panting score 3 but with the tongue protruded slightly, occasionally fully extended for short periods with excessive drooling.	Respiration rate of 120–160 breaths/min.
4	Open mouth with tongue fully extended for prolonged periods and excessive drooling; neck extended and head up.	Respiration rate greater than 160 breaths/min and may be variable due to a phase shift in respiration.
4.5	As for 4 but with the head held down; cattle ‘breath’ from flank; drooling may cease.	Variable—RR may decrease.

Panting scores were assigned based on the visual observation of respiratory dynamic and behavior, not on the estimation of the respiration rate. Source: Mader et al. [40].

**Table 4 animals-14-01914-t004:** Environmental conditions during the experimental periods.

Parameters	Season Period	
	Hot	Cool
Days for temperature and RH recording	118	122
Mean temperature (°C)	27.9	25.35
Minimum temperature (°C)	22.7	20.50
Maximum temperature (°C)	33.1	30.20
Mean relative humidity (RH%)	70.5	70.00
Minimum relative humidity (RH%)	43.00	41.00
Maximum relative humidity (RH%)	98.00	99.00
Mean daily THI	82.2	77.57
Minimum daily THI	80	76
Maximum daily THI	84	80

**Table 5 animals-14-01914-t005:** Descriptive statistics for the milk yield and milk composition parameters of 29 lactating Holstein Friesian crossbred dairy cows.

Measured Traits		Non-Genetic Parameters									
		Season		Genotype		Parity		Lactation Month		Milking Time	
Milk	Stat	Hot	Cool	HF50	HF75	2	3	2	3	AM	PM
	N	1888	1586	2156	1318	1912	1562	2518	956	1737	1737
	Mean	3.84	3.11	3.56	3.43	3.56	3.45	3.75	2.89	3.93	3.09
	Std Dev	3.11	1.37	1.43	1.30	1.54	1.16	1.48	0.79	1.47	1.15
	Min	0.5	0.20	0.20	1.30	0.2	1.3	0.20	0.50	0.3	0.2
	Max	8.8	9.0	8.80	9.0	9.0	6.4	9.0	5.80	9	7
THI	N	1888	1586	2156	1318	1912	1562	2518	956	1737	1737
	Mean	82.20	77.57	80.10	80.06	80.43	79.67	79.96	80.43	80.08	80.08
	Std Dev	0.93	1.15	2.52	2.53	2.47	2.53	2.53	2.47	2.53	2.53
	Min	80	76	76	76	76	76	76	76	76	76
	Max	84	80	84	84	84	84	84	84	84	84
Fat%	N	256	208	288	176	256	208	336	128	232	232
	Mean	2.84	2.72	2.80	2.76	2.80	2.77	2.76	2.85	2.74	2.83
	Std Dev	0.61	0.46	0.57	0.52	0.57	0.52	0.51	0.64	0.49	0.60
	Min	2.00	2.00	2.00	2.01	2.00	2.00	2.00	2.01	2.00	2.01
	Max	4.51	4.60	4.60	4.51	4.60	4.51	4.60	4.51	4.41	4.60
Prot%	N	256	208	288	176	256	208	336	128	232	232
	Mean	3.08	3.18	3.15	3.07	3.13	3.11	3.12	3.14	3.10	3.15
	Std Dev	0.32	0.30	0.30	0.33	0.30	0.34	0.31	0.32	0.35	0.27
	Min	1.84	1.65	1.85	1.65	1.84	1.65	1.65	2.02	1.65	2.33
	Max	4.11	4.62	4.62	4.11	4.62	4.11	4.62	4.11	4.62	4.21
Lact%	N	256	208	288	176	256	208	336	128	232	232
	Mean	4.61	4.76	4.73	4.60	4.70	4.66	4.66	4.72	4.64	4.72
	Std Dev	0.48	0.48	0.47	0.50	0.47	0.51	0.49	0.48	0.53	0.44
	Min	2.76	2.48	2.65	2.48	2.65	2.48	2.48	3.04	2.48	2.65
	Max	6.23	7.00	7.00	6.23	7.00	6.23	7.00	6.23	7.00	6.35
SNF%	N	256	208	288	176	256	208	336	128	232	232
	Mean	8.39	8.66	8.60	8.36	8.54	8.47	8.49	8.57	8.43	8.59
	Std Dev	0.87	0.84	0.83	0.91	0.82	0.93	0.86	0.88	0.96	0.75
	Min	5.03	4.51	5.03	4.51	5.05	4.51	4.51	5.52	4.51	6.34
	Max	11.26	12.65	12.65	11.26	12.65	11.26	12.65	11.26	12.65	11.51
Bwt	N	64	52	72	44	64	52	84	32	58	58
	Mean	356.18	328.07	351.05	331.36	355.31	329.15	346.80	335.12	343.58	343.58
	Std Dev	28.58	56.18	42.69	47.02	43.04	44.01	48.09	35.91	45.41	45.41
	Min	300	249	253	249	264	249	253	249	249	249
	Max	440	418	440	418	440	370	440	365	440	440

Stat = statistics; N = number of records; Std Dev = standard deviation; Min = minimum; Max = maximum; AM = morning; PM = afternoon; HF50 = Lactating Holstein Friesian with 50% gene level; HF75 = Lactating Holstein Friesian with 75% gene level; Fat% = fat percentages; Prot% = protein percentages; Lact% = lactose percentages; SNF = solid–not–fat percentages; THI = temperature–humidity index; Bwt = body weight of Lactating Holstein Friesian crossbred dairy cows.

**Table 6 animals-14-01914-t006:** Descriptive statistics for physiological parameters of 29 lactating Holstein Friesian crossbred dairy cows.

Measured Traits		Non-Genetic Parameters									
		Season		Genotype		Parity		Lactation Month		Milking Time	
CBT	Stat	Hot	Cool	HF50	HF75	2	3	2	3	AM	PM
	N	1888	1586	2156	1318	1912	1562	2518	956	1737	1737
	Mean	36.26	35.28	35.81	35.81	35.88	35.73	35.77	35.92	35.50	36.12
	Std Dev	0.62	0.39	0.72	0.72	0.72	0.72	0.72	0.71	0.53	0.75
	Min	34.10	32.30	32.30	32.70	32.70	32.30	32.30	33.0	32.30	32.70
	Max	39.40	37.30	39.40	39.0	39.40	38.60	39.40	38.70	38.70	39.40
RT	N	1888	1586	2156	1318	1912	1562	2518	956	1737	1737
	Mean	38.30	38.12	38.28	38.11	38.27	38.15	38.23	38.19	37.68	38.75
	Std Dev	0.77	0.73	0.73	0.78	0.73	0.78	0.74	0.80	0.48	0.58
	Min	35.50	35.70	35.70	35.60	35.60	35.70	35.60	36.20	35.60	36.80
	Max	40.60	40.80	40.80	40.50	40.80	40.80	40.80	40.80	40.0	40.80
RR	N	1888	1586	2156	1318	1912	1562	2518	956	1737	1737
	Mean	75.17	61.95	69.81	68.04	70.34	67.67	68.82	69.96	63.14	75.13
	Std Dev	7.41	10.81	11.32	11.04	10.71	11.71	11.36	10.90	9.10	9.91
	Min	60.00	36.00	36.00	40.00	36.00	40.00	36.00	40.00	36.00	40.00
	Max	88.00	88.00	88.00	88.00	88.00	88.00	88.00	88.00	88.00	88.00
PS	N	1888	1586	2156	1318	1912	1562	2518	956	1737	1737
	Mean	1.50	1.37	1.44	1.43	1.45	1.42	1.43	1.44	1.00	1.88
	Std Dev	0.5	0.48	0.49	0.49	0.49	0.49	0.49	0.49	0.06	0.32
	Min	1.00	1.00	1.00	1.00	1.00	1.002	1.00	1.00	1.00	1.00
	Max	2.00	2.00	2.00	2.00	2.00	2.00	2.00	2.00	2.00	2.00

Stat = statistics; N = number of records; Std Dev = standard deviation; Min = minimum; Max = maximum; AM = morning; PM = afternoon; HF50 = Holstein Friesian with 50% gene level; HF75 = Holstein Friesian with 75% gene level; CBT = core-body temperature; RT = rectal temperature; RR = respiration rate; PS = panting score.

**Table 7 animals-14-01914-t007:** Effect of THI, genotype, parity, and months of lactation on milk yield and milk composition parameters of 29 lactating Holstein Friesian crossbred dairy cows.

Non-Genetic Parameters		Milk Yield and Composition Parameters				
		Milk (L/d)	Fat %	Protein %	Lactose %	SNF %
THI	76–78	2.25 ± 0.03 ^a^	2.72 ± 0.04 ^a^	3.17 ± 0.02 ^a^	4.74 ± 0.04 ^a^	8.62 ± 0.07 ^a^
	79–81	3.46 ± 0.04 ^b^	2.64 ± 0.05 ^b^	3.13 ± 0.02 ^a^	4.69 ± 0.04 ^a^	8.51 ± 0.07 ^a^
	82–84	3.51 ± 0.04 ^b^	2.98 ± 0.06 ^c^	3.06 ± 0.03 ^b^	4.61 ± 0.03 ^b^	8.34 ± 0.08 ^b^
*p*-value		˂0.05	˂0.05	˂0.05	<0.05	<0.05
Genotype	HF50	3.03 ± 0.04 ^a^	2.78 ± 0.03 ^a^	3.16 ± 0.02 ^a^	4.75 ± 0.03 ^a^	8.62 ± 0.06 ^a^
	HF75	3.55 ± 0.04 ^b^	2.72 ± 0.04 ^a^	3.08 ± 0.02 ^b^	4.61 ± 0.03 ^b^	8.37 ± 0.06 ^b^
*p*-value		˂0.05	˂0.05	<0.05	<0.05	<0.05
Parity	2	3.33 ± 0.03 ^a^	2.78 ± 0.03 ^a^	3.15 ± 0.02 ^a^	4.72 ± 0.03 ^a^	8.57 ± 0.05 ^a^
	3	3.24 ± 0.03 ^b^	2.80 ± 0.04 ^a^	3.12 ± 0.02 ^a^	4.67 ± 0.04 ^b^	8.48 ± 0.06 ^a^
*p*-value		<0.05	˂0.05	˂0.05	˂0.05	˂0.05
Months of lactation	2	3.62 ± 0.03 ^a^	2.69 ± 0.04 ^a^	3.09 ± 0.02 ^a^	4.64 ± 0.02 ^a^	8.41 ± 0.06 ^a^
	3	3.19 ± 0.04 ^b^	2.81 ± 0.05 ^b^	3.15 ± 0.02 ^b^	4.73 ± 0.04 ^b^	8.58 ± 0.08 ^b^
*p*-value		<0.05	<0.05	<0.05	<0.05	<0.05

Means on the same column with different superscript letters are significantly different (p ≤ 0.05); THI = temperature–humidity index; HF50 = lactating Holstein Friesian dairy cows with 50% gene level; HF75 = lactating Holstein Friesian dairy cows with 75% gene level; milk yield; fat percentage, protein percentage; lactose percentage; SNF = solids–not–fat percentage.

**Table 8 animals-14-01914-t008:** Effect of THI, genotype, parity, and months of lactation on the physiological parameters of 29 lactating Holstein Friesian crossbred dairy cows.

Non-Genetic Parameters		Physiological Parameters			
		CBT (°C)	RT (°C)	RR (breaths/min)	PS
THI	76–78	35.16 ± 0.03 ^a^	38.04 ± 0.02 ^a^	59.74 ± 0.30 ^a^	1.31 ± 0.01 ^a^
	79–81	35.32 ± 0.01 ^a^	38.15 ± 0.02 ^b^	69.79 ± 0.34 ^b^	1.46 ± 0.01 ^b^
	82–84	36.24 ± 0.01 ^b^	38.27 ± 0.02 ^b^	72.78 ± 0.29 ^c^	1.46 ± 0.01 ^b^
*p*-value		˂0.05	<0.05	˂0.05	˂0.05
Genotype	HF50	35.57 ± 0.02 ^a^	38.21 ± 0.02 ^a^	64.95 ± 0.28 ^a^	1.37 ± 0.01 ^a^
	HF75	35.57 ± 0.02 ^a^	38.06 ± 0.02 ^b^	63.34 ± 0.28 ^b^	1.36 ± 0.01 ^b^
*p*-value		˂0.05	˂0.05	˂0.05	˂0.05
Parity	2	35.82 ± 0.01 ^a^	38.25 ± 0.01 ^a^	69.39 ± 0.21 ^a^	1.45 ± 0.01 ^a^
	3	35.79 ± 0.03 ^a^	38.15 ± 0.03 ^b^	68.55 ± 0.24 ^b^	1.43 ± 0.01 ^a^
*p*-value		<0.05	<0.05	<0.05	˂0.05
Months of lactation	2	35.54 ± 0.01 ^a^	38.07 ± 0.02 ^a^	66.14 ± 0.30 ^a^	1.34 ± 0.01 ^a^
	3	35.60 ± 0.02 ^b^	38.19 ± 0.03 ^b^	68.73 ± 0.32 ^b^	1.36 ± 0.02 ^a^
*p*-value		<0.05	<0.05	<0.05	<0.05

Means on the same column with different superscript letters are significantly different (*p* ≤ 0.05); THI = temperature–humidity index; HF50 = lactating Holstein Friesian crossbreds dairy cows with 50% gene level; HF75 = lactating Holstein Friesian crossbreds dairy cows with 75% gene level; CBT = core body temperature; RT = rectal temperature RR = respiratory rate; PS = panting score.

**Table 9 animals-14-01914-t009:** Effect of the interaction of the genotype and THI on the milk yield, milk composition, and physiological parameters of 29 lactating Holstein Friesian crossbred dairy cows.

Genotype	HF50			HF75			*p*-Value
THI Class	76–78	79–81	82–84	76–78	79–81	82–84	
Milk yield (L/d)	2.68 ± 0.08 ^a^	2.81 ± 0.05 ^a^	3.58 ± 0.04 ^b^	3.53 ± 0.10 ^b^	3.68 ± 0.05 ^c^	3.45 ± 0.05 ^b^	<0.05
Fat %	2.74 ± 0.10 ^a^	2.86 ± 0.06 ^b^	2.84 ± 0.04 ^b^	2.53 ± 0.12 ^c^	2.74 ± 0.07 ^a^	2.83 ± 0.05 ^b^	<0.05
Protein %	3.26 ± 0.05 ^a^	3.27 ± 0.28 ^a^	3.11 ± 0.02 ^b^	3.03 ± 0.06 ^c^	3.08 ± 0.04 ^c^	3.09 ± 0.03 ^c^	<0.05
Lactose %	4.89 ± 0.08 ^a^	4.90 ± 0.05 ^a^	4.67 ± 0.03 ^b^	4.53 ± 0.10 ^c^	4.61 ± 0.06 ^b^	4.63 ± 0.04 ^b^	<0.05
SNF %	8.88 ± 0.15 ^a^	8.92 ± 0.10 ^a^	8.49 ± 0.06 ^b^	8.25 ± 0.19 ^c^	8.38 ± 0.12 ^b^	8.42 ± 0.08 ^b^	<0.05
CBT (°C)	35.22 ± 0.04 ^a^	35.34 ± 0.02 ^a^	36.10 ± 0.02 ^b^	35.23 ± 0.05 ^a^	35.37 ± 0.02 ^a^	36.14 ± 0.02 ^b^	<0.05
RT (°C)	38.07 ± 0.06 ^a^	38.07 ± 0.03 ^a^	38.20 ± 0.03 ^b^	38.14 ± 0.07 ^c^	38.11 ± 0.03 ^c^	38.12 ± 0.03 ^c^	<0.05
RR (breaths/min)	55.51 ± 0.70 ^a^	62.97 ± 0.41 ^b^	72.72 ± 0.40 ^c^	55.73 ± 0.83 ^a^	62.29 ± 0.43 ^b^	72.52 ± 0.41 ^c^	<0.05
PS	1.23 ± 0.04 ^a^	1.38 ± 0.02 ^b^	1.44 ± 0.02 ^c^	1.21 ± 0.04 ^a^	1.37 ± 0.02 ^b^	1.47 ± 0.02 ^c^	<0.05

Means on the same column with different superscript letters are significantly different (*p* ≤ 0.05); THI = temperature–humidity index; HF50 = lactating Holstein Friesian crossbred dairy cows with 50% gene level; HF75 = lactating Holstein Friesian crossbred dairy cows with 75% gene level; milk yield; fat percentage, protein percentage; lactose percentage; SNF = solids–not–fat percentage; CBT = core body temperature (°C); RT = rectal temperature (°C); RR = respiratory rate (breaths/min); PS = panting score.

**Table 10 animals-14-01914-t010:** Effect of parity and THI on the milk yield, milk composition, and physiological parameters of 29 lactating Holstein Friesian crossbred dairy cows.

Parity	Second Parity			Third Parity			*p*-Value
THI	76–78	79–81	82–84	76–78	79–81	82–84	
Milk	2.78 ± 0.05 ^a^	3.52 ± 0.05 ^b^	3.68 ± 0.04 ^c^	2.84 ± 0.05 ^d^	3.25 ± 0.06 ^b^	3.60 ± 0.05 ^c^	<0.05
Fat %	2.75 ± 0.06 ^a^	2.67 ± 0.05 ^a^	2.98 ± 0.04 ^b^	2.73 ± 0.05 ^a^	2.64 ± 0.06 ^a^	3.03 ± 0.07 ^b^	<0.05
Protein %	3.24 ± 0.03 ^a^	3.15 ± 0.04 ^b^	3.07 ± 0.03 ^c^	3.13 ± 0.03 ^b^	3.14 ± 0.04 ^b^	3.11 ± 0.03 ^b^	<0.05
Lactose %	4.84 ± 0.06 ^a^	4.72 ± 0.05 ^b^	4.61 ± 0.04 ^c^	4.68 ± 0.05 ^c^	4.70 ± 0.06 ^b^	4.65 ± 0.06 ^c^	<0.05
SNF %	8.82 ± 0.10 ^a^	8.58 ± 0.09 ^b^	8.37 ± 0.08 ^c^	8.50 ± 0.09 ^b^	8.54 ± 0.11 ^b^	8.46 ± 0.11 ^c^	<0.05
CBT (°C)	35.27 ± 0.02 ^a^	35.85 ± 0.02 ^b^	36.33 ± 0.01 ^c^	35.23 ± 0.02 ^a^	35.77 ± 0.02 ^b^	36.37 ± 0.02 ^c^	<0.05
RT (°C)	38.18 ± 0.03 ^a^	38.21 ± 0.03 ^a^	38.34 ± 0.02 ^b^	37.95 ± 0.03 ^a^	38.19 ± 0.03 ^a^	38.31 ± 0.03 ^b^	<0.05
RR	60.41 ± 0.39 ^a^	72.23 ± 0.39 ^b^	75.52 ± 0.30 ^c^	59.07 ± 0.36 ^a^	71.09 ± 0.43 ^b^	75.50 ± 0.41 ^c^	<0.05
PS	1.35 ± 0.02 ^a^	1.49 ± 0.02 ^b^	1.50 ± 0.01 ^b^	1.31 ± 0.01 ^a^	1.48 ± 0.02 ^b^	1.50 ± 0.02 ^b^	<0.05

Means on the same column with different superscript letters are significantly different (*p* ≤ 0.05); THI = temperature–humidity index; milk yield in liters per day; fat percentage, protein percentage; lactose percentage; SNF = solids–not–fat percentage; CBT = core body temperature (°C); RT = rectal temperature (°C); RR = respiratory rate (breaths/min); PS = panting score.

**Table 11 animals-14-01914-t011:** Effect of months of lactation and THI on milk yield, milk composition, and physiological parameters of 29 lactating Holstein Friesian crossbred dairy cows.

Lactation Month	Second Month of Lactation			Third Month of Lactation			*p*-Value
THI	76–78	79–81	82–84	76–78	79–81	82–84	
Milk yield (L/d)	3.33 ± 0.07 ^a^	3.38 ± 0.04 ^a^	3.87 ± 0.05 ^b^	2.98 ± 0.13 ^c^	3.25 ± 0.07 ^a^	3.18 ± 0.06 ^a^	<0.05
Fat %	2.62 ± 0.08 ^a^	2.78 ± 0.05 ^b^	2.78 ± 0.03 ^b^	2.66 ± 0.15 ^a^	2.79 ± 0.10 ^b^	2.90 ± 0.05 ^c^	<0.05
Protein %	3.15 ± 0.05 ^a^	3.19 ± 0.03 ^a^	3.07 ± 0.02 ^b^	3.18 ± 0.09 ^a^	3.15 ± 0.05 ^a^	3.14 ± 0.03 ^a^	<0.05
Lactose %	4.71 ± 0.07 ^a^	4.77 ± 0.04 ^a^	4.60 ± 0.03 ^b^	4.75 ± 0.14 ^a^	4.72 ± 0.08 ^a^	4.72 ± 0.05 ^a^	<0.05
SNF %	8.57 ± 0.13 ^a^	8.69 ± 0.08 ^b^	8.36 ± 0.06 ^c^	8.64 ± 0.25 ^b^	8.55 ± 0.15 ^a^	8.57 ± 0.09 ^a^	<0.05
CBT (°C)	35.15 ± 0.03 ^a^	35.29 ± 0.02 ^a^	36.03 ± 0.02 ^b^	35.32 ± 0.06 ^a^	35.40 ± 0.03 ^a^	36.21 ± 0.02 ^b^	<0.05
RT (°C)	38.04 ± 0.05 ^a^	38.03 ± 0.03 ^a^	38.14 ± 0.03 ^b^	38.20 ± 0.09 ^c^	38.18 ± 0.05 ^b^	38.17 ± 0.03 ^b^	<0.05
RR (breaths/min)	54.37 ± 0.62 ^a^	61.34 ± 0.37 ^b^	71.47 ± 0.42 ^c^	56.51 ± 1.10 ^a^	64.20 ± 0.60 ^b^	73.66 ± 0.41 ^c^	<0.05
PS	1.21 ± 0.03 ^a^	1.37 ± 0.02 ^b^	1.44 ± 0.02 ^c^	1.22 ± 0.06 ^a^	1.37 ± 0.03 ^b^	1.48 ± 0.02 ^c^	<0.05

Means on the same column with different superscript letters are significantly different (*p* ≤ 0.05); THI = temperature–humidity index; milk yield in liters per day; fat percentage, protein percentage; lactose percentage; SNF = solids–not–fat percentage; CBT = core body temperature (°C); RT = rectal temperature (°C); RR = respiratory rate (breaths/min); PS = panting score.

**Table 12 animals-14-01914-t012:** Pearson correlation coefficients between THI, milk yield, and milk composition parameters of 29 lactating Holstein Friesian crossbred dairy cows.

Milk Yield and Composition Parameters	THI	Fat %	Prot %	Lact %
Milk yield	0.24023 *			
Fat %	0.15598 **			
Prot %	−0.15029 **	0.19051 *		
Lact %	−0.13537 **	0.23410 *	0.98623 *	
SNF %	−0.14132 **	0.22388 *	0.99625 *	0.98957 *

* *p* < 0.0001; ** *p* ≤ 0.05; THI = temperature–humidity index; milk yield in liters per day; fat percentage, protein percentage; lactose percentage; SNF = solids–not–fat percentage.

**Table 13 animals-14-01914-t013:** Pearson correlation coefficients between THI and physiological parameters of 29 lactating Holstein Friesian crossbred dairy cows.

Physiological Parameters	THI	CBT	RT	RR
CBT	0.67457 *			
RT	0.63497 *	0.52908 *		
RR	0.63497 *	0.72192 *	0.66772 *	
PS	0.16427 *	0.52930 *	0.70487 *	0.69245 *

* *p* < 0.0001; THI = temperature–humidity index; CBT = core body temperature (°C); RT = rectal temperature (°C); RR = respiratory rate (breaths/min); PS = panting score.

## Data Availability

The data are available on request due to restrictions.

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
