# Peer review of "Heat Stress Effects on Physiological and Milk Yield Traits of Lactating Holstein Friesian Crossbreds Reared in Tanga Region, Tanzania"

_animals, 2024, doi:10.3390/ani14131914_

Round 1
Reviewer 1 Report
Comments and Suggestions for Authors
row 19 solid – not - fat percentages - Delete the milk protein and lactose or delete "solid – not - fat percentages" and add the missing components
row 35 replace blood with genes
row 101 insufficiently clear sentence
row 170 since they are crossbreeds with the Holstein breed, do not use the term Holstein cattle
row 186 replace blood with genes
row 246-250 Why did you use this distribution THI into classes? Can you back up this allocation with a reference? Here you can see how THI is divided into classes in the temperate continental climate zone https://hrcak.srce.hr/file/411303
row 296 (table 5) In the part related to THI classes and milk yield per day, do you have an explanation for these results?
row 410 Do you have an explanation for these results?
Comments on the Quality of English Language
The English language used in the paper is correct and understandable, but it should definitely be reviewed by a native speaker.
Author Response
Dear Reviewer 1,
I have addressed the comments as required.
Kindly find attached the responses per each comment given. After addressing the comments as highlighted in the manuscript. Prof. Gota Morota extensively edited the English as you suggested.
I look forward to receiving your positive feedback .
Best regards,
Vincent

Reviewer 2 Report
Comments and Suggestions for Authors
Heat stress in cattle and its negative impact on the health, reproduction and milk productivity of cows have been described in numerous publications, including original and review works.
Since the problem of heat stress in dairy herds is constantly present in the world, both science and practice are forced to look for new analytical and remedial methods in this area.
The presented manuscript shows analytical examples enabling the assessment of correlations of many physiological parameters, milking and milk composition in different groups of cows in a herd kept in a region with extremely high average annual temperature and low milk yield in cows.
The presented correlation results may have practical significance in modern management, monitoring and analysis of cattle herds in terms of welfare, health and milk production not only in hot regions of the world, but also in other regions where seasonally during summer, cows become exposed to this serious problem. environmental and milk production.
The presented results on a small research group may constitute an individual reference point for other dairy herds in terms of the quantity and quality of milk raw material, which translates into the economics of this production, as well as the quality of food products of the dairy industry in the situation of constant warming of the climate also recorded on other continents. .
However, the manuscript contains some inaccuracies listed below, which require clarification and correction or addition.
The animals studied are lactating cows, and the text often uses the term "dairy cattle" or "hybrids" in many sections, which does not necessarily mean such a cow. This also applies to section titles, table titles and the title of the manuscript itself!? This must be taken into account.
The animals studied in the manuscript are lactating cows, and the text often uses the term "dairy cattle" or "hybrids" in many sections, which does not necessarily mean such a cow. This also applies to section titles, table titles and the title of the manuscript itself!? This must be taken into account.
The material section does not describe important herd parameters such as the number of cows in the herd, average milk yield per lactation, average cow weight, epidemiological and parasitological status of the herd, type of barn, type of technical and milking equipment, and pasture area. This needs to be completed.
The individual tables do not show the number of animals analyzed (n), especially since the methodology section divides the cows into groups tested in the cold and warm seasons. The tables should indicate whether all animals in both seasons were considered together or separately. This also needs to be corrected and supplemented.
The manuscript contains a multi-directional statistical analysis based on a sufficient number of animals, which has publication value for cattle science and practice. However, using for detailed statistical analysis subgroups of only a few cows giving a few data (e.g. n= 3 dairy cattle in the 4th parity), as well as comparing these results in different sections of the manuscript (phrases in lines 359-371 and Table 8), and the next sections and conclusions (lines 724-727) is debatable and may be questioned. The number "n" below 6 usually does not allow for a reliable statistical analysis of research work. Therefore, the analysis, discussion and formulation of final conclusions in relation to subgroups with small sample sizes that may generate doubts should be verified in these appropriate places in the manuscript.
The authors of the manuscript should also be careful about stylistic simplifications in the text, and so in the first sentence of the conclusions section (lines 716-717):
“The results of this study show that daily milk yield and milk composition decline while physiological parameters (CBT, RT, RR and PS) increase at THI thresholds ranging between 79 and 84.”
It is better to add a more adequate and precise phrase:
“The results of this study show that daily milk yield and the values ​​​​of the tested milk composition parameters declined while physiological parameters (CBT, RT, RR and PS) increased at THI thresholds ranging between 79 and 84.”
Comments on the Quality of English Language
Noted above
Author Response
Dear Reviewer 2,
I have addressed all the comments as you suggested. The comments have greatly improved the manuscript. Kindly, find attached the responses per each comments in the Table below and the changes are highlighted in the manuscript. After addressing the comments, Prof. Gota Morota extensively Edited the English as you recommended.
I look forward to your positive feedback.
Best regards,
Vincent

Round 2
Reviewer 2 Report
Comments and Suggestions for Authors
The manuscript has currently revised and improved according to the reviewer's previous recommendations.